# Dimensionality-Transformed Remote Sensing Data Application to Map Soil Salinization at Lowlands of the Syr Darya River

Kanat Samarkhanov [1,2,3,†], Jilili Abuduwaili [1,2,3,*,†], Alim Samat [1,2,3,†], Yongxiao Ge [1,2,3,†], Wen Liu [1,2,3,†], Long Ma [1,2,3,†], Zhassulan Smanov [1,2,3,4,†], Gabit Adamin [4,†], Azamat Yershibul [4,†] and Zhassulan Sadykov [5,†]

1   State Key Laboratory of Desert and Oasis Ecology, Xinjiang Institute of Ecology and Geography, Chinese Academy of Sciences, Urumqi 830011, China
2   Research Center for Ecology and Environment of Central Asia, Chinese Academy of Sciences, Urumqi 830011, China
3   University of Chinese Academy of Sciences, Beijing 100049, China
4   U.U.Uspanov Kazakh Research Institute of Soil Science and Agrochemistry, Almaty 050060, Kazakhstan
5   Social Work and Tourism Department, Faculty of Applied Sciences and the Department of Postgraduate Education, Esil University, Nur-Sultan 010005, Kazakhstan
*   Correspondence: jilil@ms.xjb.ac.cn
†   These authors contributed equally to this work.

**Abstract:** The problem of saving soil resources and their reclamation measures under current climate change conditions attracts the world community's close attention. It is relevant in the Syr Darya River's lowlands, where the secondary soil salinization processes have intensified. The demand for robust methods to assess soil salinity is high, and the primary purpose of this study was to develop a quantitative analysis method for soil salinity estimation. We found a correspondence between the sum of salts in a topsoil layer to the Landsat 8 data in the Tasseled cap transformation of the image values. After testing several methods, we built a prediction model. The K-nearest neighborhood (KNN) model with a coefficient of determination equal to 0.96 using selected predictors proved to be the most appropriate for soil salinity assessment. We also performed a quantitative assessment of soil salinity. A significant increase in a salt-affected area and the mean soil sum expressing an intensification of secondary soil salinization from 2018 to 2021 was found. The increasing temperature values, decreasing soil moisture, and agricultural use affect the extension of salt-affected ground areas in the study area. Thus, the soil moisture trend in the Qazaly irrigation zone is negative and declining, with the highest peaks in early spring. The maximum temperature has a mean value of 15.6 °C (minimum = −15.1 °C, maximum = 37.4 °C) with an increasing trend. These parameters are evidence of climate change that also affects soil salinization. PCA transformation of the Landsat-8 satellite images helped to remove redundant spectral information from multiband datasets and map soil salinity more precisely. This approach simultaneously extends mapping opportunities involving visible and invisible bands and results in a smaller dataset.

**Keywords:** soil salinity; prediction; secondary salinization; trend; temperature; soil moisture; climate change; heating; aridization

## 1. Introduction

Healthy soil maintenance is the key to effective agricultural crop production, and the decline in soil productivity is affected by land degradation. Soil salinization is one of the significant land degradation types caused by natural and human-induced effects at global and local levels [1–3]. About one-third of global cultivated land is affected by salinization in over 1 billion hectares in over 100 countries [4–6]. Climate change is often reported to be an indirect driver of soil, and groundwater salinity [7–9]. Long-time observations help us understand the influence of climatic fluctuations on different variables because of

the ability to decompose complex time series into effects driven by different factors with methods such as the Mann–Kendall test [10–13].

In this regard, monitoring soil salinity is essential to land management. Some researchers consider soil texture mapping, while others describe mapping the soil electric conductivity measured by electromagnetic-sensitive tools [14–16]. However, field-based soil salinity monitoring is limited in spatial and temporal dimensions. Satellite remote sensing is a standard method to evaluate the extent and severity of soil salinity and its effects. Thus, satellite-measured information is practical for soil salinity evaluation in terms of quantity and quality [17,18].

Many studies describe the use of optical remote sensing data to assess or map soil salinity in bare soils [19–35]. The era of open data supplies the end-users with space images of high resolution, such as Landsat-8 and Sentinel-2, widely used in environmental research and soil salinity assessment [36–43].

Most are based on band ratios, and different soil salinity indices have been formulated, becoming a universal method to derive soil salinity information. Another approach is aggregating multiple bands using dimensionality reduction techniques, such as the principal component analysis (PCA) [44]. PCA application increases data interpretability with a minimum information loss. The Tasseled Cap Transformation (TCT) is a universal and simple yet powerful tool to analyze information on arid and semiarid lands [45].

Satellite monitoring was used as a robust method for soil salinity assessment in Kazakhstan [46–48]. Since proper soil salinization tracking is critical for global food security, remote sensing is essential for assessing soil salinity, primarily because of a lack of data in the Syr Darya River's Kazakh section.

Some researchers considered aerial geologic mapping, biological variability, soil fertility, water supply, Aral crisis consequences, human health, land use, soil moisture, soil reclamation, leaching of saline land, soil salinity, and soil salinity effects on crops and natural vegetation in their soil salinity mapping projects for the Aral sea surroundings in Kazakhstan.

Problems of soil salinity in Central Asia became well-known after the breakup of the USSR [49]. Soil salinity issues became more specific, and an international topic [50–54]. From the 2000s, the Kazakh Institute of Soil Sciences and Agricultural Chemistry experts contributed to soil salinity studies and salinization in southern and southeastern Kazakhstan [55,56].

In the 2010s, soil salinity studies were conducted at the national level and supported by international publications reported by Issanova et al. [57]. Factors in intensifying degradation processes were identified and consisted of poor irrigation systems, losses in irrigation water, rising levels of mineralized groundwater, worsening of irrigation water, and salinity and alkalinity of soil [58].

Some of the considered publications reported using spectral indices to monitor soil salinization [59]. New studies are devoted to the driving forces in the Aral Sea region [60,61]. This study utilized the Tasseled Cap Transformed images and climate variable trends to assess the salinization of soils in the Kazakh part of the Syr Darya River basin.

## 2. Materials and Methods

### 2.1. Description of the Study Area

The Qazaly (Kazaly) irrigation zone (QIZ) is situated in the southwestern part of Kazakhstan and covers the Syr Darya (Syrdarya) River's delta. The Syr Darya River runs through four countries of Central Asia that are in the center of Eurasia. It has an area of about 15,000 km$^2$ and geographic coordinates between 45°–46.2° N and 60.8°–62.7° E (Figure 1).

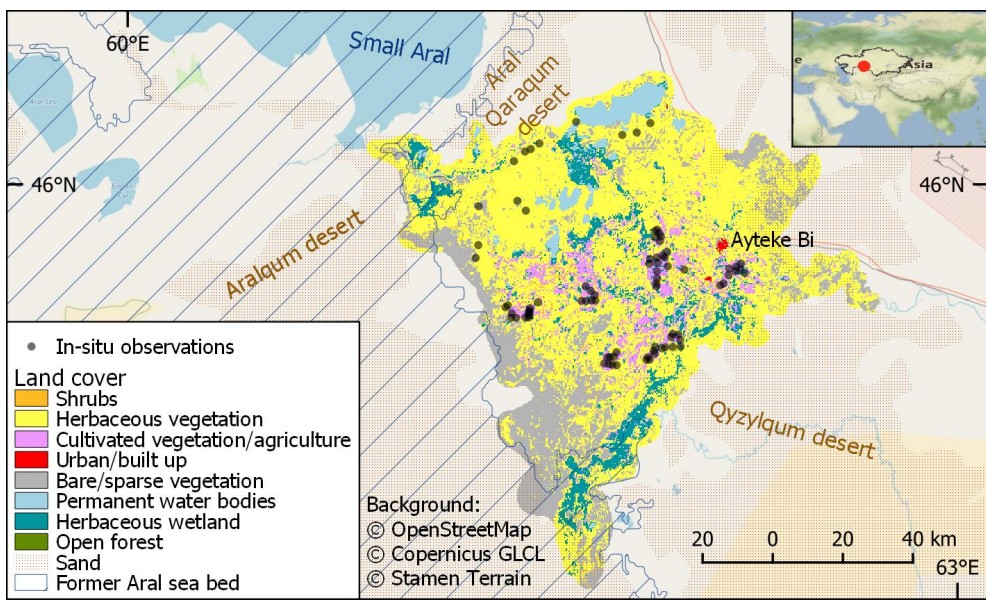

**Figure 1.** The Qazaly irrigation zone and sampling sites.

The study site is within the Turan lowland near the former Aral Sea. The area is mainly occupied by the Qazaly (Kazaly) irrigation zone and is surrounded by sands such as the vast Qyzylqum (Kyzylkum) Desert and the recently formed Aralqum (Aralkum) Desert. It is a flat lowland with an elevation below 100 m, with arid climatic conditions where the annual precipitation is below 200 mm [62]. The vegetation period here starts in March and ends in October [63]. Actively used croplands marked with pink areas on the map were chosen to assess soil salinity as essential for agricultural land use.

### 2.2. The Data and Research Methods

The research workflow consisted of significant steps illustrated in Figure 2. Work started with preprocessing the satellite images (extracting transformed indices) and checking for in situ data consistency. Then, the correlation between soil parameters and transformed indices was studied, followed by proper data selection, and a machine learning algorithm was applied to monitor the soil salinity. The derived time series were decomposed at the final stage, and soil salinization trends were calculated (detailed description follows).

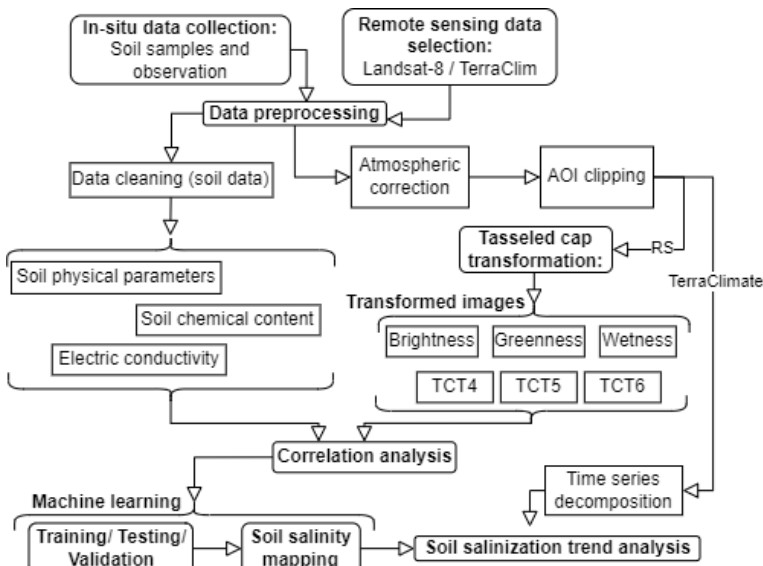

**Figure 2.** The workflow of the study.

1. Field research was conducted once in May 2018, including soil sampling, land cover, and land use observation at 83 locations. The 93 soil samples of the topsoil layer (0–20 cm) and samples at 20–50 cm and 50–100 cm depth were collected (30 samples at each), and soil EC was measured. The coordinates (WGS-84) were measured using the Garmin T650 hand-held global positioning system and a mobile phone with an accuracy of about 10 m due to locations situated primarily in distant areas without a mobile network. The field measurement of the soil electric conductivity (Soil EC) was carried out using the STEP Combi 3000+ device (Figure 3). Pictures of landscapes and vegetation were taken at each location. The field measurements included the soil temperature and moisture measurements using the portable STEP Combi 5000 tool (Germany).

2. Data consisting of 14 parameters (sand content (%, Sa.), clay content (%, Cl.), soil organic matter (%, OM-Humus), electric conductivity (dS m$^{-1}$, EC$_{soil}$), soil temperature (°C, T$^o_{soil}$), soil moisture (mm, Moist$_{soil}$), a sum of salt in the soil (g kg$^{-1}$, SoS), chlorine (Cl$^-$), sulfate (SO$_4^{2-}$), calcium (Ca$^{2+}$), magnesium (Mg$^{2+}$), sodium (Na$^+$), and potassium (K$^+$)) were prepossessed to test possible interconnections of variables and to find predictors of the soil salinity expressed by soil electric conductivity. The soil moisture was selected as a potential predictor for the soil salinity, and its correlation with the satellite remote sensing data was tested afterward. The coefficient of determination for selected predictors in the sum of salt in the soil of the medium (20–50 cm) and the lowest (50–100 cm) soil layers did not exceed 0.6; data on the two layers were not included in further modeling. The coordinates of points were checked accurately, and data refinement resulted in a topsoil layer dataset consisting of variables on 31 points (Table 1) out of the initial 93. The sum of salts contains the sum of ions and is included in the table.

3. For the physical and chemical analysis, soil samples were transported to the U.U.Uspanov Kazakh Research Institute of Soil Science and Agrochemistry (Almaty, Kazakhstan) for further laboratory analysis. The analysis included several soil parameters: the water hood, mechanical content analysis, electric conductivity, and soil humus content. The soil electrical conductivity was measured using electrodes in an aqueous solution (with distilled water) with a 1:5 ratio (soil: water).

4. The Pearson's coefficient was calculated to characterize the connections between measured parameters according to the equation below.

$$r_{xy} = \frac{\sum_{i=1}^{n}(x_i - \bar{x})(y_i - \bar{y})}{\sqrt{\sum_{i=1}^{n}(x_i - \bar{x})^2 \sum_{i=1}^{n}(y_i - \bar{y})^2}}, \tag{1}$$

where $n$ is sample size, $x_i$, $y_i$ are the individual sample points indexed with i, $\bar{x}$, and $\bar{y}$ is the sample means that is estimated as follows:

$$\bar{x} = \frac{1}{n}\sum_{i=1}^{n} x_i, \tag{2}$$

where $\bar{x}$ is mean value for all values of $x$, and the equation for $y$ is the same.

5. The,n statistical distribution was assessed using the box plot method and the observed and derived data (soil EC, temperature, moisture, and mechanical content at three horizons) [64,65].

6. Soil texture was classified by soil mechanical content data derived after the laboratory analysis according to the USDA classification, and its repetition was also analyzed [66]. The open-source Python software tool Soiltexture 1.0.4 (https://pypi.org/project/soiltexture/ (accessed on 21 September 2022)) was applied for this purpose. To visualize the soil clusters for the study area, the texture triangle [67] implemented through the Python tool SoilTriangle (https://github.com/mishagrol/SoilTriangle (accessed on 21 September 2022)) was used. In the next step, the distribution of the

sum of salt in the soil and other parameters within soil texture classes were checked. Soil salt chemical content analysis was also performed to describe the soil salinization.

**Table 1.** The soil samples and measured parameters.

| Name | E | N | Sa. | Si. | Cl. | OM | T_s | Moist_s | EC_s | SoS |
|------|-----|-----|------|-----|------|------|------|---------|------|------|
| S103 | 45.68 | 61.4 | 34.5 | 18 | 47.5 | 0.97 | 32.9 | 8.6 | 0.49 | 16.77 |
| S108 | 45.68 | 61.4 | 34 | 23 | 43 | 1.56 | 39.3 | 15.6 | 5.58 | 17.01 |
| S113 | 45.68 | 61.4 | 17 | 30 | 53 | 1.18 | 42 | 8.6 | 1.23 | 11.66 |
| S134 | 45.9 | 61.3 | 44 | 0 | 56 | 1.11 | 39.7 | 14.2 | 3.51 | 28.98 |
| S143 | 45.7 | 61.6 | 20 | 20 | 60 | 1.52 | 35.8 | 14.5 | 3.33 | 16.12 |
| S144 | 45.5 | 61.7 | 20 | 28 | 52 | 1.32 | 34 | 7.4 | 1.74 | 15.02 |
| S146 | 45.5 | 61.7 | 23.5 | 18 | 58.5 | 0.93 | 37.5 | 10.4 | 2.96 | 28.6 |
| S147 | 45.5 | 61.8 | 39 | 10 | 51 | 1.04 | 40.5 | 18.1 | 8.51 | 52.03 |
| S149 | 45.6 | 61.7 | 21.5 | 23 | 55.5 | 1.04 | 36.8 | 9.1 | 4.72 | 29.83 |
| S150 | 45.6 | 61.7 | 32 | 25 | 43 | 1.11 | 39.8 | 7.2 | 1.58 | 11.02 |
| S151 | 45.6 | 61.8 | 33.5 | 19 | 47.5 | 1.11 | 36 | 5.1 | 0.42 | 6.17 |
| S152 | 45.6 | 61.9 | 31.5 | 11 | 57.5 | 0.45 | 32.7 | 19.7 | 4.93 | 42.41 |
| S154 | 45.6 | 61.9 | 23 | 15 | 62 | 0.97 | 37.1 | 16.0 | 3.53 | 8.11 |
| S157 | 45.6 | 61.9 | 15 | 23 | 62 | 1 | 33.5 | 9.6 | 1.43 | 9.35 |
| S158 | 45.6 | 61.9 | 36.5 | 11 | 52.5 | 0.41 | 33.2 | 10.8 | 2.62 | 19.67 |
| S160 | 45.6 | 62.0 | 40.5 | 12 | 47.5 | 0.79 | 33.1 | 7.6 | 0.77 | 11.6 |
| S163 | 45.62 | 62.0 | 26 | 13 | 61 | 1.92 | 30.2 | 7.0 | 1.15 | 10.42 |
| S165 | 45.61 | 62.0 | 41.5 | 11 | 47.5 | 0.51 | 27 | 26.6 | 9.91 | 36.52 |
| S167 | 45.8 | 62.0 | 27.5 | 24 | 48.5 | 1.41 | 31.8 | 7.2 | 0.86 | 7.94 |
| S171 | 45.82 | 62.0 | 57 | 17 | 26 | 1.23 | 39.5 | 3.8 | 0.56 | 7.24 |
| S172 | 45.82 | 62.0 | 45 | 21 | 34 | 1.68 | 40.4 | 10.2 | 1.77 | 21.07 |
| S174 | 45.8 | 62.0 | 46 | 15 | 39 | 0.93 | 35.3 | 16.0 | 7.12 | 27.83 |
| S180 | 45.9 | 62.0 | 36.5 | 14 | 49.5 | 0.65 | 39 | 11.8 | 1.22 | 13.31 |
| S181 | 45.9 | 62.0 | 29 | 21 | 50 | 0.65 | 39.8 | 18.4 | 5.12 | 29.37 |
| S183 | 45.8 | 62.0 | 51.5 | 15 | 33.5 | 0.79 | 41.8 | 5.2 | 0.51 | 6.43 |
| S184 | 45.9 | 62.0 | 27.5 | 23 | 49.5 | 1.44 | 41.9 | 12.9 | 4.13 | 37.22 |
| S187 | 45.8 | 62.2 | 21 | 26 | 53 | 1.13 | 45.1 | 6.6 | 2.95 | 9.73 |
| S188 | 45.8 | 62.2 | 53.5 | 10 | 36.5 | 0.38 | 42.7 | 22.6 | 8.61 | 36.13 |
| S189 | 45.8 | 62.2 | 39 | 11 | 50 | 0.75 | 42.7 | 29.9 | 10.87 | 52.24 |
| S191 | 45.8 | 62.2 | 57 | 12 | 31 | 1.06 | 41.3 | 21.6 | 7.68 | 25.08 |
| S192 | 45.7 | 62.1 | 44 | 21 | 35 | 1.89 | 38.9 | 13.1 | 1.01 | 10.31 |

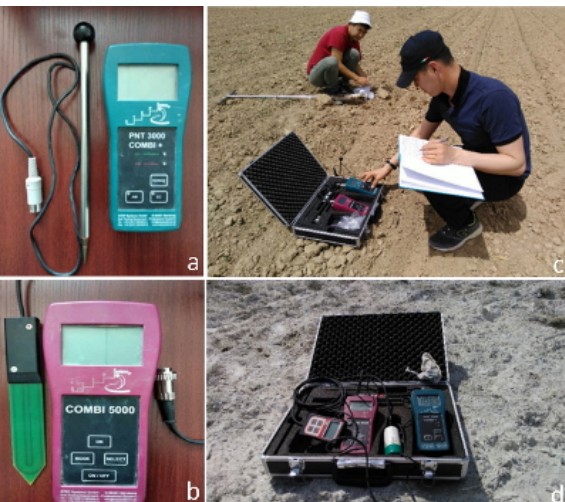

**Figure 3.** In-situ measurements in 2018: (**a**) STEP PNT 3000+ Combi—electric conductometer; (**b**) STEP Combi 5000—for soil temperature and moisture measurements; (**c**) soil EC measurements and soil sampling; (**d**) soil moisture, temperature measurements (co-authors of this article Zh. Smanov and A. Yershibul are collecting soil samples).

7.  Tasseled cap transformation was applied to Landsat-8 bands, and 6 coefficients (placed in 6 bands) that improved reflectance of vegetation, soil, and water: greenness, wetness, and brightness, were obtained as coefficients from 4 to 6. These coefficients successfully differentiate soil from vegetation, and in this study, it was decided to apply them to obtain soil information. The values were proposed by Baig et al. [68]. The single scene acquired by Landsat-8 (Collection 2 Tier 1 calibrated top-of-atmosphere (TOA) reflectance) was selected for the Qazaly cropping zone to test the possible relationships between the soil salinity and Landsat 8 bands acquired on 13 May 2018 (path 160, row 028). The Google Earth Engine (https://developers.google.com/earth-engine/guides/arrays_array_images (accessed on 20 September 2022)) web service was used to derive the Tasseled Cap Transformed coefficients. Correlation tests of derived TCT coefficients and soil salinity features of the study area were used to define best-matching predictors.

8.  The soil salinity prediction modeling was built after selecting its predictors. The input dataset, consisting of variables on 31 samples, was established. Due to the high variance of other measurements, the sum of salts in the soil was used. Images acquired on 13 May 2018, 16 July 2018, 2 September 2018, 18 September 2018, and 20 October 2018 were selected for modeling.

    Three machine learning techniques were chosen to analyze the observed sum of salt in the soil: multiple linear regression (MLR), polynomial regression, K-nearest neighborhood (KNN), and the partial least squares regression (PLSR) for this purpose [35,69,70]. Conditions of using a small yet valuable dataset were similar: the share of train and test values was 70% and 30% within 31 samples. Machine learning open-source Scikit-learn software was used for a regression [71].

    After this comparison, applying the K-nearest neighbor model to predict soil salinity using Landsat-8 images was decided. It is based on $k$ nearest neighbor for input points, is calculated using the distance between values as the weight in this research, and is expressed using Equation (3):

$$d(x,y) = \sqrt{\sum_{i=1}^{n}(x_i - y_i)^2},\tag{3}$$

    where $x$ is an input band value, and $y$ is the sum of salt in the soil.

9.  The quantitative soil salinity assessment for 2013–2021 was performed by applying the regression K-nearest neighbor model. The pixel-wise sum of salt in the soil was derived over 42 periods and calculated for the study area's croplands. Then, the areas affected by soil salinity were measured and added to the table. Data on the sum of salt in the soil was organized as a set of time series. Trends and seasonal variations of derived parameters were calculated using an additive decomposition model via Python Statsmodel tool (https://www.statsmodels.org/stable/index.html (accessed on 27 September 2022)). The additive model can be described by Equation (4).

$$y(t) = Level + Trend + Seasonality + Noise,\tag{4}$$

    where *Level* is an average value, *Trend*—increasing/decreasing value, *Seasonality*—short-term cycle in the series, *Noise*—is a random variation, and their summary $y$ is a model of time series.

    The primary purpose of this study was to set up the proper method and data to quantify soil salinity for the Qazaly irrigation zone using commonly applied satellite image data, and the results are described in the next part.

10. To monitor the difference in climate change trends, we considered the soil moisture and maximum temperature data by TerraClimate between July 2002 and December 2021 [72]. TerraClimate is a global dataset of the monthly surface climate and water balance. It uses the interpolation method to calculate the data. Thus, we applied the Mann–Kendall test to derive the trends in the soil moisture and maximum temperature

at 2 different points for comparison: the first located at the study area (N45.591, E61.9065) and the second at the upper part down to the southern region in the border zone between Kazakhstan and Uzbekistan (N40.88593, E68.092595). Points were selected due to the soil salinity conditions based on the field studies in 2018. We applied the results on the salt-affected soil areas from the previous part to compare the climatic parameters and found several concurrences for the Qazaly irrigation zone.

## 3. Results

### 3.1. Statistic Processing of the Field Data

Laboratory analysis of the soil samples conducted at the U.U.Uspanov Kazakh Research Institute of Soil Science and geochemistry and field observations (2018) resulted in the table with 14 parameters at 3 depths: sand content (%, Sa.), clay content (%, Cl.), soil organic matter (%, OM-Humus), electric conductivity (dS m$^{-1}$, EC$_{soil}$), soil temperature (°C, T$^o_{soil}$), soil moisture (mm, Moist$_{soil}$), a sum of salt in the soil (g kg$^{-1}$, SoS), chlorine (Cl$^-$), sulfate (SO$_4^{2-}$), calcium (Ca$^{2+}$), magnesium (Mg$^{2+}$), sodium (Na$^+$), and potassium (K$^+$).

The correlation coefficients were calculated from the input field data and plotted using the "heat" map method (Figure 4).

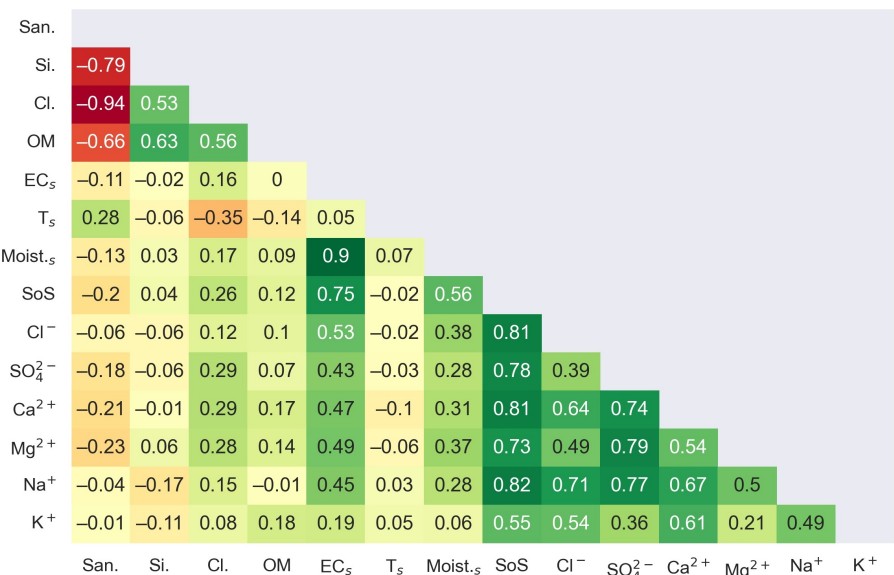

**Figure 4.** The correlation matrix for the different soil parameters at the study area.

A valuable strong positive correlation was observed between the soil electric conductivity (EC$_{soil}$, dS m$^{-1}$) and soil moisture (Moist$_{soil}$,%).

The soil texture classes, including sand, clay, and silt, strongly correlate negatively. The chemical composition of soils demonstrated a high positive correlation between the sum of salt in the soil (SoS, g kg$^{-1}$), chlorine (Cl$^-$), g kg$^{-1}$ (0.81), sulfates-SO$_4^{2-}$, g kg$^{-1}$ (0.78), calcium-Ca$^{2+}$, g kg$^{-1}$ (0.81), magnesium (Mg$^{2+}$), g kg$^{-1}$ (0.73), and sodium-Na$^+$, g kg$^{-1}$ (0.82) as shown in the right corner of the plot.

The correlation between soil chemical content and other parameters was insufficient, so a correlation between field-observed soil electric conductivity and soil moisture was examined. Though it showed a strong relationship, no correlation between them and TCT-image pixel values was found to compare these two parameters and Landsat 8 Tasseled Cap Transformed data. Furthermore, collected data on the sum of salt in the soil were considered to assess the soil salinity as the most proper. Then, a statistical distribution was plotted for the sum of salt in the soil (Figure 5).

The box visualized the distribution of salts at the topsoil layer starting from minimum values above 0, then with a median of 15.71 g kg$^{-1}$, Q2 of 29.6 g kg$^{-1}$ reaching 62.89 g kg$^{-1}$ at the highest value. In addition, some outlier values were observed with high values

(67.77–79.32 g kg$^{-1}$). The sum of salt in the soil decreased at the medium layer and ranged between 2.49 and 21 g kg$^{-1}$ with a median of 10.28 g kg$^{-1}$. Its sum increased within a range of 1.95–17.54 g kg$^{-1}$ and a median of 9.55 g kg$^{-1}$. We could notice that the principal sum of salt in the soil is concentrated on the soil surface.

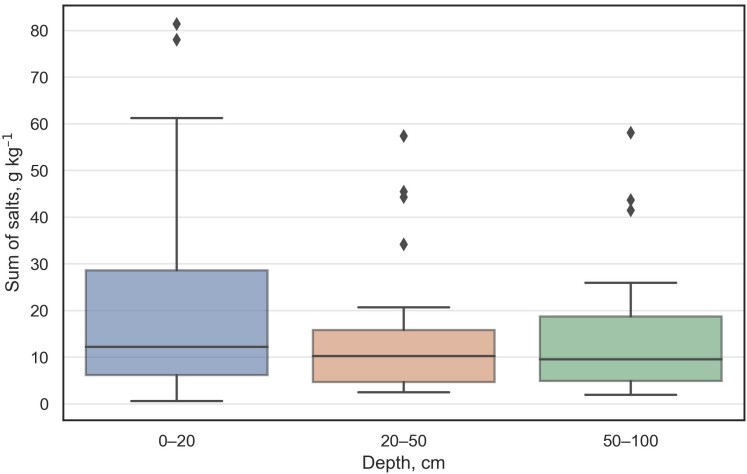

**Figure 5.** The statistical distribution of the sum of salt in the soil (g kg$^{-1}$) at 3 soil layers.

### 3.2. Distribution of Soil Texture and Salt Chemical Composition

The soil data were classified according to USDA classification by soil texture [66], and its repetition was also analyzed. It was found that most soil samples at three horizons mostly contain clay (23–79%) represented by 54 samples for the topsoil layer and by 15 samples at the medium one and 20 at the underneath layer according to Table 2.

**Table 2.** Number of samples by soil classes according to USDA soil texture classification.

| Soil Mech. Content | Soil Layer Depth, cm | | | |
|---|---|---|---|---|
| | **0–20** | **50–100** | **20–50** | **Total** |
| loamy sand | 1 | | | 1 |
| sandy loam | 4 | 4 | 1 | 9 |
| sandy clay loam | 26 | 6 | 5 | 37 |
| sandy clay | 6 | 3 | 2 | 11 |
| clay loam | 1 | 2 | 2 | 5 |
| clay | 54 | 15 | 20 | 89 |
| silty clay | 1 | | | 1 |

Clay content demonstrated the maximum between 14–95% (with a median clay content at 49.5%) at the topsoil layer (0–20 cm) and decreased at the medium soil layer (20–50 cm) with values from 19% to 92%, while the lowest values of 23–92% were observed at the soil layer between 50–100 cm. The soil moisture was distributed in the reverse direction with 0.12–29.9% values at the top layer followed by 4.1–37.02% at the medium layer and increasing in the lowest layer (4.74–40.17%).

The sandy clay loam follows clay (contains 45–72.5% of sand), sandy clay (contains 45.5–56% of sand), sandy loam (contains 66–79.5% of sand), clay loam (contains 35–39.5% of clay) and loamy sand (contains 85.5% of sand), and silty clay (contains 41% of silt) from top to down layers. The sand, silt, and clay share were plotted on the soil texture triangle to analyze their distribution graphically (Figure 6).

The dots are positioned due to the share of particles, while their size is a gradient of soil moisture, and their color expresses the range of soil electric conductivity measured in situ. The distribution of particles by soil texture classes demonstrates the concentration of the clay in the upper part of the triangle for all layers. The soil moisture has the lowest

values at the topsoil layer (0.12–29.87%) compared to the other two layers. At the medium soil layer, moisture has a higher range with values between 4.1% and 37.02%, while the layer underneath its variation is within 4.74–40.17%.

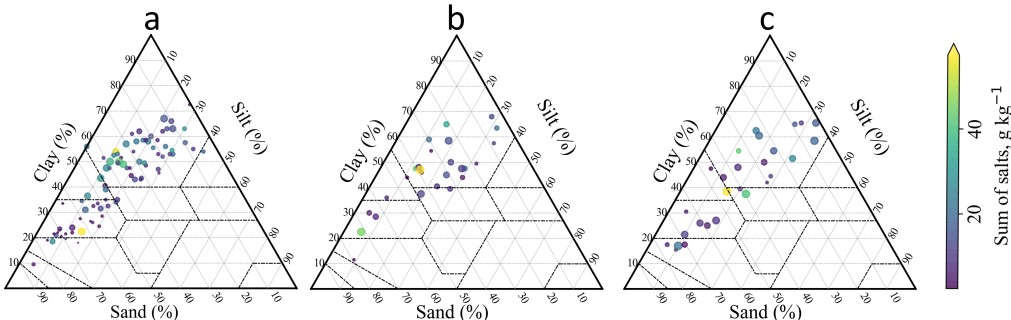

**Figure 6.** The soil texture triangles for 3 soil layers: (**a**) 0–20 cm; (**b**) 20–50 cm; (**c**) 50–100 cm; dot size—soil moisture; dot color—soil EC.

The plot shows that the noticeable sum of salt in the soil mostly corresponds to the highest values in a clay zone at three layers in the upper part of the triangles. The common property for three soil layers was revealed; they were mostly higher in the 10–30% silt-containing samples. The relationship between soil texture and soil salinity was not found in the images during this study.

Therefore, the soil salt chemical content was also considered in parallel with soil texture data. The distribution of major anions and cations by soil layers was also analyzed according to Table 3.

**Table 3.** Total share salt chemical composition (major ions) for 3 soil layers.

|  | $SO_4^{2-}$ | $Cl^-$ | Total |
|---|---|---|---|
| $Ca^2$ | 111 | 6 | 117 |
| $Na^+$ | 25 | 11 | 36 |
| Total | 136 | 17 | 153 |

Thus, the topsoil layer contained mostly $Ca^2$ (63 samples), $SO_4^{2-}$ (89 samples), less $Na^+$ (30 samples), and $Cl^-$ (4 samples). $Na^+$ and $SO_4^{2+}$ prevailed in two underlying layers. There were other ions, including $Mg^{2+}$ and $K^+$, their quantity was not negligible. Thus, according to the salt chemical content analysis, the significant sum of salt in the soil in the study area mainly consists of calcium sulfate ($CaSO_4$, dry gypsum) and sodium sulfate ($Na_2SO_4$, Glauber's salt) and a few samples contained sodium chloride (NaCl, table salt).

According to the degree of soil salinization (non-saline soil: <1 g kg$^{-1}$; mildly saline soil: 1–6 g kg$^{-1}$; moderately saline soil: 6–10 g kg$^{-1}$; severely saline soil: 10–20 g kg$^{-1}$, and saline soil: >20 g kg$^{-1}$) a preliminary evaluation of salinization was made. Based on the analysis results, the soils at 83 sampling sites demonstrated different degrees of salinity: saline (39%), severely saline (25%), moderately saline (19%), mildly saline (16%), and non-saline (1%).

### 3.3. Selection of Predictors for Soil Salinity Assessment

The preliminary analysis of the spatial distribution of topsoil salts was performed on the image acquired on 13 May 2018. It was used because of date correspondence for site visiting and image acquisition. The Landsat 8 RGB image was considered, revealing a significant positive relationship between red and blue bands and the sum of salt in the soil according to Figure 7. For these bands, the coefficient of determination is equal to 0.77 and 0.79, followed by the blue band (0.74). The NIR band demonstrated a relatively weak coefficient (0.43), while SWIR 1 and SWIR 2 bands were significantly correlated with the sum of salt in the soil (0.7 each).

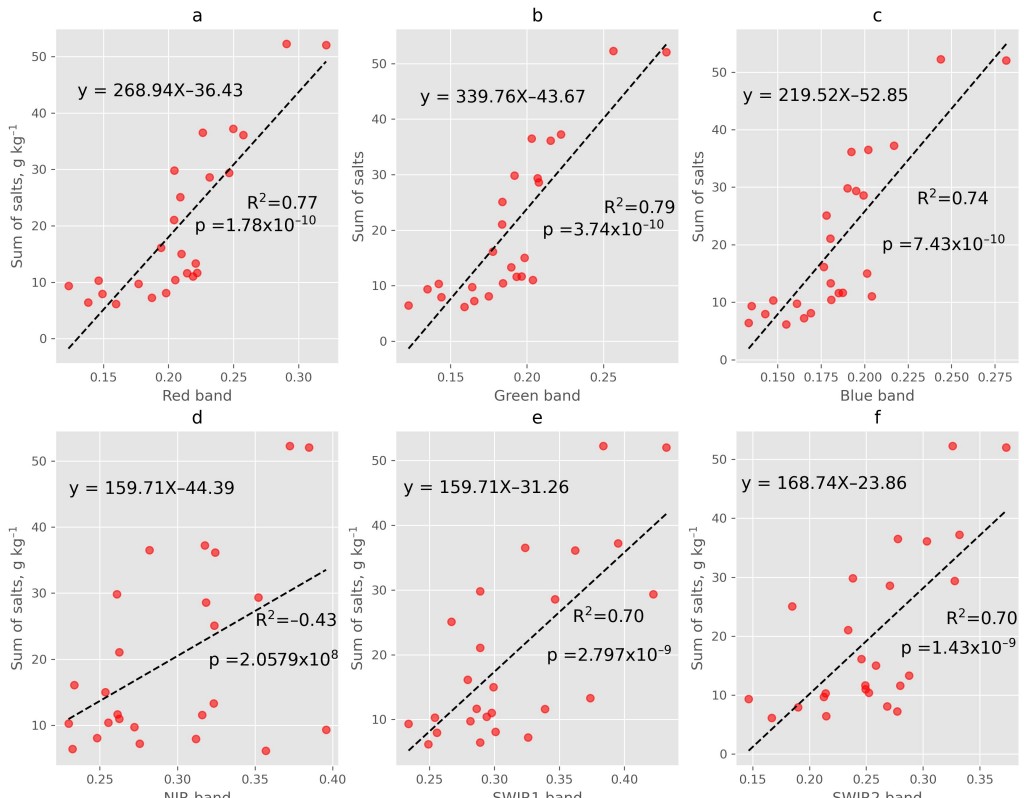

**Figure 7.** Linear the fit of the sum of salt in the soil and the Landsat-8 RGB values at the topsoil layer: (**a**) red band; (**b**) green band; (**c**) blue band; (**d**) NIR band; (**e**) SWIR 1 band; (**f**) SWIR 2 band.

Then, the satellite image bands were processed by applying TCT coefficients (Table 4), as explained in the Section 2.

**Table 4.** TCT coefficients for Landsat 8 at-satellite reflectance.

| LandSat-8 TCT | (Blue) Band 2 | (Green) Band 3 | Red Band 4 | NIR Band 5 | SWIR1 Band 6 | SWIR2 Band 7 |
|---|---|---|---|---|---|---|
| Brightness | 0.3029 | 0.2786 | 0.4733 | 0.5599 | 0.508 | 0.1872 |
| Greenness | −0.2941 | −0.243 | −0.5424 | 0.7276 | 0.0713 | −0.1608 |
| Wetness | 0.1511 | 0.1973 | 0.3283 | 0.3407 | −0.7117 | −0.4559 |
| TCT4 | −0.8239 | 0.0849 | 0.4396 | −0.058 | 0.2013 | −0.2773 |
| TCT5 | −0.3294 | 0.0557 | 0.1056 | 0.1855 | −0.4349 | 0.8085 |
| TCT6 | 0.1079 | −0.9023 | 0.4119 | 0.0575 | −0.0259 | 0.0252 |

It was found that topsoil bare salts corresponded to pixels with the highest values in the brightness band of the TCT image (Tasseled Cap Transformed image). The sum of salt in the soil was selected as a variable directly indicating the soil salinity, among other variables derived after the field sampling. Some observation sites were covered with white crusts of salts lying on the top of the soil, as shown in the image (Figure 8).

Therefore, they were selected to analyze a relationship between spectral reflectance at sampling points and values at bare soil pixels of a TCT image. Some were excluded from analysis due to their inconsistency due to different wetness conditions that affected their spectral reflectance; on the other hand, a sampling distance of 30 m concerning the positioning of points led to the planimetric shifting of pixels reflecting salt. The standard variance contributed to the GPS's positioning accuracy around 5–10 m [73].

The linear models were developed to check the relationships between the sum of salt in the soil (dependent variable) and predictors for 31 points according to Table 1: brightness, greenness, wetness, TCT4, TCT5, and TCT6 and plotted (Figure 9).

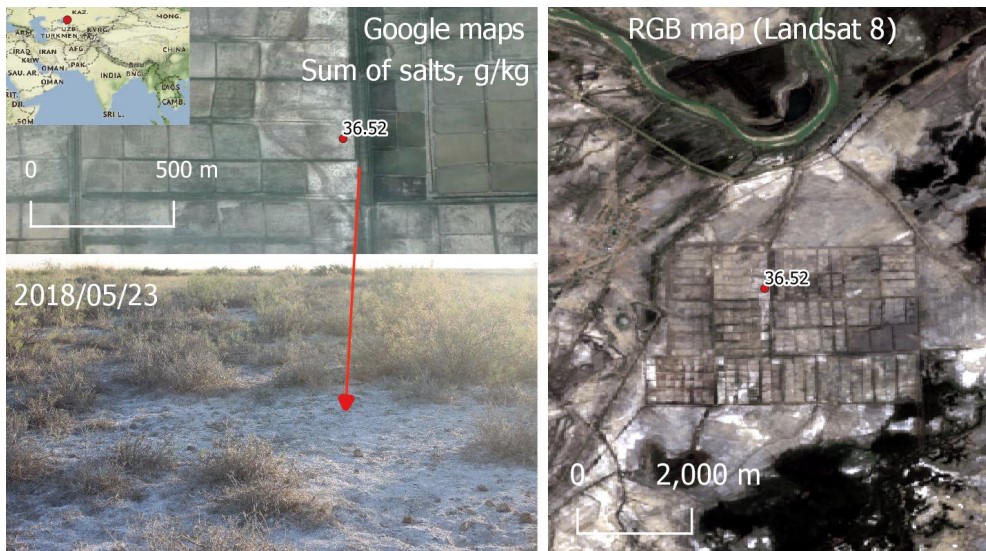

**Figure 8.** An example of a site with a saline crust and its reflection on Google maps, Landsat-8 RGB image (13 May 2018), and on-site photography (2018-05-23). Coordinates of the point: N45.61574, E61.96481.

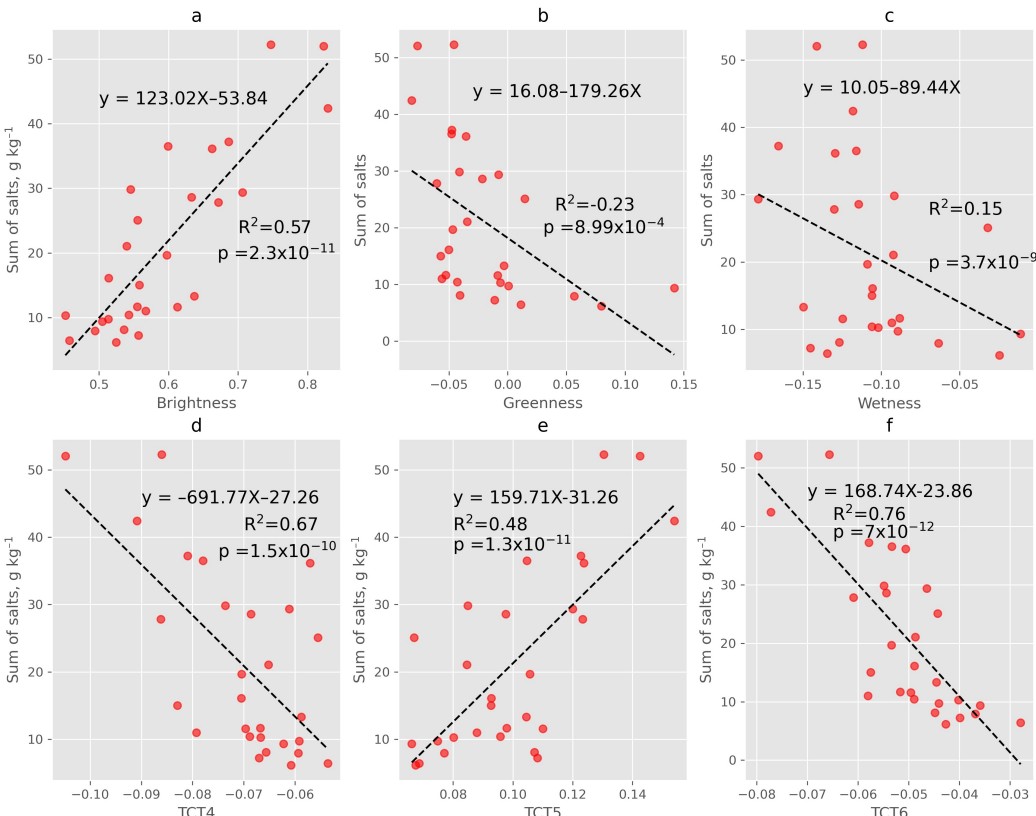

**Figure 9.** Linear fit of the sum of salt in the soil and the Landsat-8 Tasseled Cap Transformed image bands values at the topsoil layer: (**a**) brightness; (**b**) greenness; (**c**) wetness; (**d**) TCT4; (**e**) TCT5; (**f**) TCT6.

The sum of salt in the soil is considered a dependent variable. Coefficients of determination expressing the relationship between predictors and a soil sum of salt were derived. It demonstrates a significantly strong correlation between the sum of salt in the soil and brightness (positive), TCT4 (negative), and TCT6 (negative), while greenness, wetness, and TCT5 have little hostile relation to it.

Considering the results of this analysis, red bands, green bands, TCT4 bands, and TCT 6 bands highly correlated to the salinity were selected for further modeling and building a prediction model for the sum of salt in the soil.

### 3.4. The Soil Salinity Modeling

After selecting soil salinity predictors, the dataset consisting of 31 point samples with variables on the sum of salt in the soil was used to build models for soil salinity prediction on the satellite image. Conditions of using a small yet valuable dataset were similar: the train and test values share was 70% and 30% in 31 samples. Predictive modeling results can be described by regression scores given in Table 5.

**Table 5.** The soil salinity prediction modeling scores.

| Regression Score | Regression Model | | |
|---|---|---|---|
| | **PLSR** | **KNN** | **MLR** |
| Coefficient of determination | 0.98 | 0.96 | 0.94 |
| Root mean squared error, g kg$^{-1}$ | 10.06 | 7.73 | 9.01 |

The highest correlation value was derived by KNN modeling, the lowest by PLSR, the medium by MLR, and the lowest root means squared error of 7.73 g kg$^{-1}$ is obtained for the KNN modeling. The last one is the best-matching one among the three applied models. The predicted sum of salt in the soil and the initial salt amount derived after the laboratory analysis are also visualized graphically by plotting for better understanding (Figure 10).

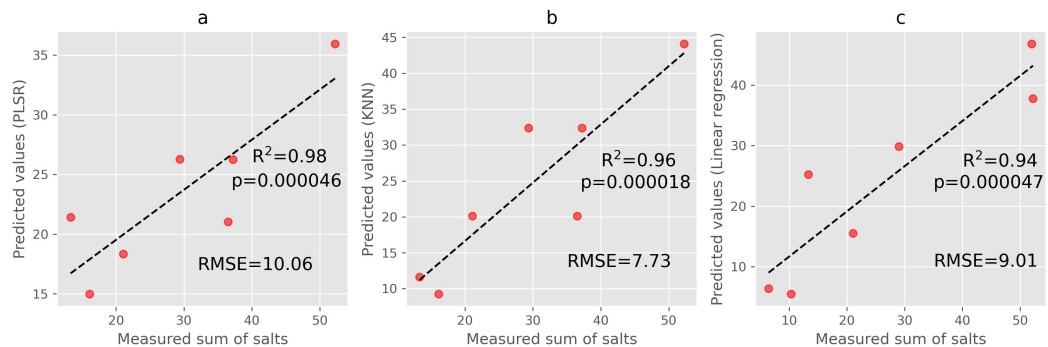

**Figure 10.** Linear fit of the soil salinity prediction modeling results for the topsoil layer: (**a**) PLSR; (**b**) KNN; (**c**) MLR.

The highest variation is observed in PLSR-predicted data; MLR-modeled data has a higher correlation, while the best linear fit was found in the plot of the KNN model. After this comparison, the K-nearest neighbor model was used to predict soil salinity using Landsat-8 images. It is based on $k$ nearest neighbor for input points and is calculated using the distance between values as the weight in this research and is expressed using Equation (3):

$$d(x, y) = \sqrt{\sum_{i=1}^{n}(x_i - y_i)^2},$$ (5)

where $x$ is an input band value, and $y$ is sum of salt in the soil .

### 3.5. The Soil Salinization at Croplands

The sum of salt in the soil was predicted for croplands in the study area using cloudless Landsat-8 images (path 160, row 028) acquired on 42 periods from 2013 to 2021, covering spring, summer, and autumn. The soil salinity maps were plotted for start and end spring observation dates to demonstrate the spatial extents according to Figure 11.

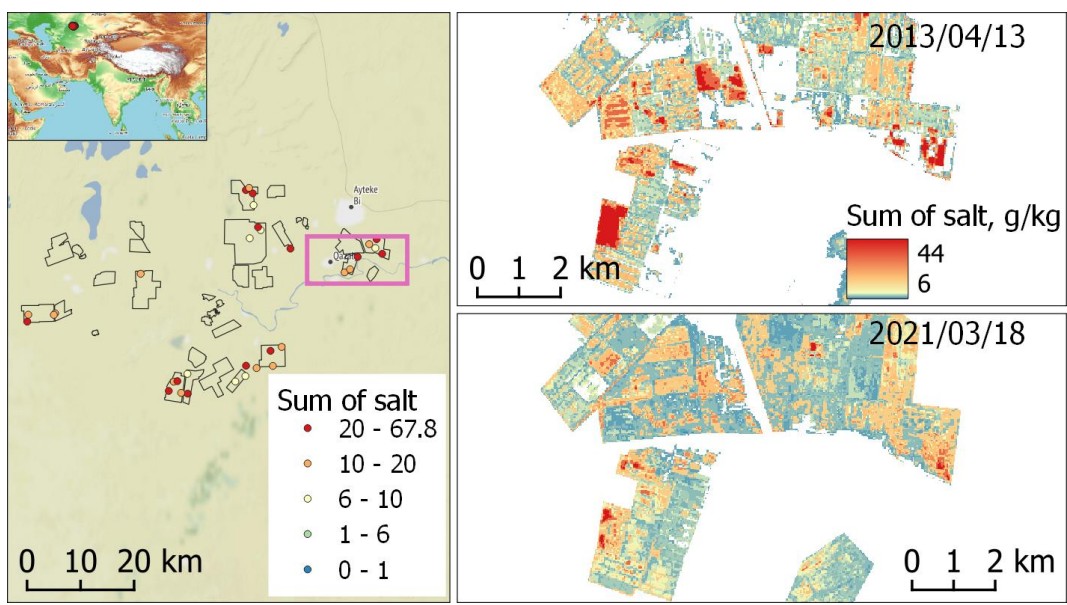

**Figure 11.** Soil salinity in 2 periods for the topsoil layer in the study area.

The sample croplands demonstrated a large saline hot spot area on 13 April 2013 and its significantly shrunk extent on 18 March 2021 with extremely high values (>20 g kg$^{-1}$). The smaller number of pixels with salt-affected soils in an older image and its increase in the recent image can be noticed.

All pixels with a derived value over 6 g kg$^{-1}$ were considered salt-affected soils. Then, the dynamics of the calculated sum of salt in the soil (Figure 12a) and salt-affected soils (Figure 12b) with their trends were plotted to make conclusions on the soil salinization for the study area.

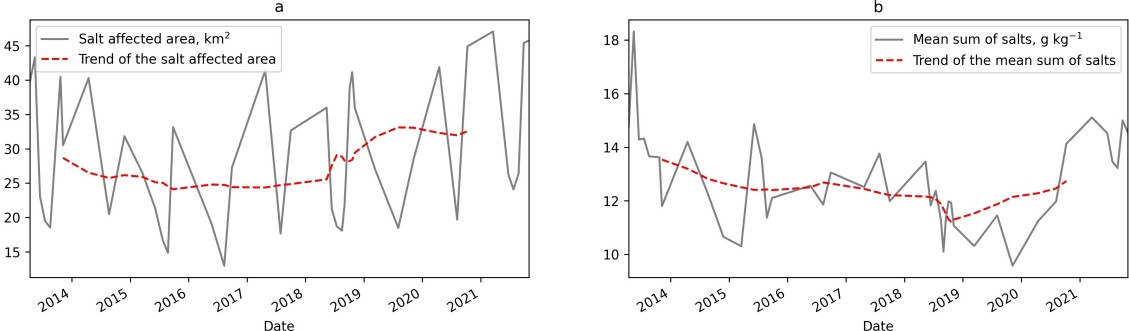

**Figure 12.** Dynamics of the sum of salt in the soil in croplands for the topsoil layer: (**a**) salt-affected croplands (km$^2$); (**b**) mean sum of salt in the soil (g kg$^{-1}$).

Figure 12 shows the seasonal variance of the salt-affected soils from 2013 to 2021. The maximum amounts of salt registered the two highest peaks for each year in spring (April–May) and autumn (October–November), with a range between 31 km$^2$ (26 November 2014) and 47 km$^2$ (18 March 2016). The lowest peaks were recorded in summer (July–August), with a range between 13 km$^2$ (11 August 2014) and 30.5 km$^2$ (11 July 2013). Its trend demonstrates a decrease in the area from 26.5 km$^2$ (16 April 2014) to 24 km$^2$ (26 September 2015) and an increase up to 33.1 km$^2$ (4 August 2019).

Figure 12b shows the temporal variance of the mean sum of salt in the soil from 2013 to 2021. The maximum amounts of salt were registered in spring with a range between 9.6 and 18.3 g kg$^{-1}$. There are two highest peaks on the plot corresponding to 15 May 2013 (18.3 g kg$^{-1}$) and 6 June 2015 (14.9 g kg$^{-1}$). The lowest sum of salt in the soil was observed

on 18 March 2015 (10.3 g kg$^{-1}$) and 11 August 2019 (9.6 g kg$^{-1}$). A straight increase is recorded up to 15.11 g kg$^{-1}$ on 18 March 2021.

### 3.6. Soil Salinity and Climate Change

The Mann–Kendall test derived the trends in soil moisture and maximum temperature (Figure 13). The linear trend of the soil moisture trend at the Qazaly irrigation zone (Point 1 according to the Materials and Methods part of this paper) is negative, and it is declining, despite the short-term increase. As for the values of soil moisture, they were below 20 mm per day, with the highest peaks in March (minimum = 0.9 mm, mean = 3.6 mm, maximum = 18.1 mm). Point 2 in the upper lowlands of Syr Darya demonstrated a maximum of 89.4 mm (minimum = 4.8 mm, mean = 29.3 mm), which is much higher related to water supply conditions. We can conclude that in our study area, the water scarcity is higher; and therefore, it also affects the salinization due to declination of the soil moisture long-term trend.

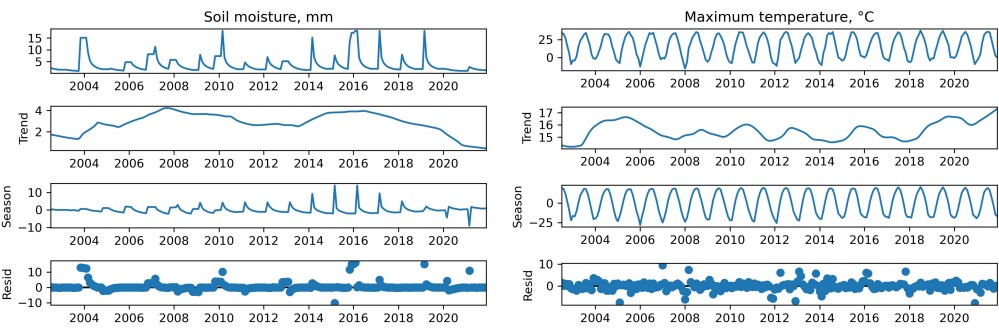

**Figure 13.** Dynamics of the soil moisture and the maximum temperature at Qazaly irrigation zone.

We could see an increase in the trend line regarding the maximum temperature. At the study area (Point 1) mean value for maximum temperature was 15.6 °C (minimum = −15.1 °C, maximum = 37.4 °C). At Point 2, temperature shows higher values with peaks in July and August. The mean value for maximum temperature was 21.8 °C (minimum = −3.6 °C, maximum = 38.8 °C). Therefore, we observe the heating in dry lands that also affects other aspects, including soil salinization. We considered the relationship between climate heating and salt-affected soil areas according to the figure (Figure 14).

Similarities between the soil moisture, temperature, and salt-affected ground area patterns are shown in the plot. We conclude that the relationship between soil salinization and climate change is vital.

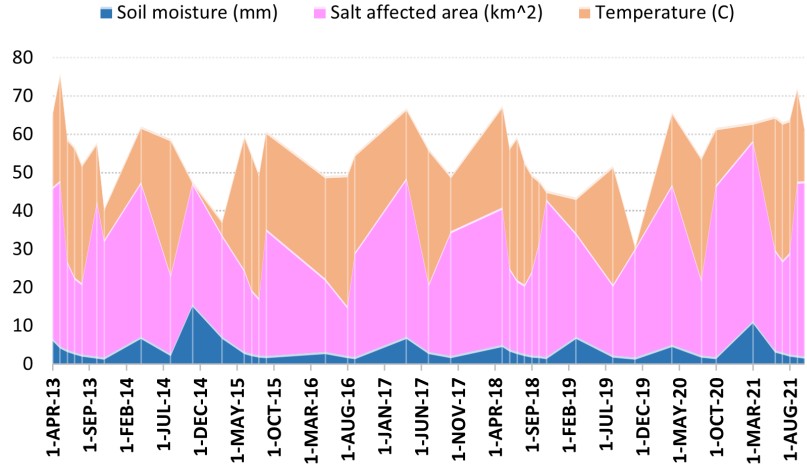

**Figure 14.** Dynamics of the soil moisture, temperature and salt-affected areas in QIZ.

## 4. Discussion

It was indicated in this paper that the correlation between soil chemical content and other parameters needed to be increased. In contrast, the soil electric conductivity and soil moisture were the most correlated field data. The correlation between soil texture and salinity did not guarantee the correlation between soil texture and image bands due to its coverage by salts at sampling points applied for the analysis.

Though researchers often report proving connections between soil electric conductivity and image bands, no correlation was found between soil electric conductivity and spectral band records in this study. PCA-like transformed images helped reveal a significant correlation between the sum of salt in the soil and Landsat TCT-corrected band images. It can be explained by the correspondence of selected in situ data to areas with significant salt amounts on the surface. Yet "Tasseled Cap" images combined with normal Landsat 8 bands helped separate bare ground at some areas and salt though they had high values in the brightness TCT-band. Same time soil salinity is not necessarily directly reflected in spectral bands in wet and veobtainated lands.

Derived soil salinity fluctuations for the study area with an intensification of soil salinization from 2018 to 2021 could be caused by changes in soil water content that influence seasonal soil salinity values. Coinciding increases in a salt-affected area and the soil sum of salt values can demonstrate a significant increasing trend of the soil secondary salinization in the study area.

## 5. Conclusions

The results of this research work are essential and may be applied to maintain soil health and agricultural production balance. Therefore, the study's primary purpose was to establish a quantitative analysis method for soil salinity estimation and to have a sense of soil salinization in the study area. The technique used open satellite data to make it applicable to analog sites.

Coinciding increases in a salt-affected area and the mean soil sum of salt demonstrated a significant trend of soil secondary salinization in the study area from 2018 to 2021. This leads to more stressful conditions for agricultural production and ecologic equilibrium. Therefore, it is necessary to develop an environmental management plan.

The increasing temperature with decreasing soil moisture, along with the agricultural use, leads to an extension of salt-affected ground areas in the study area. This can be a topic for further research. The study's results can contribute to solving the problem of soil resource conservation and reclamation under the current climate change conditions.

**Author Contributions:** Conceptualization, K.S. and J.A.; methodology, K.S., A.S., Y.G., W.L. and L.M.; software, K.S.; validation, K.S., A.S., W.L. and Z.S. (Zhassulan Smanov); formal analysis, G.A. and A.Y.; investigation, Z.S. (Zhassulan Sadykov); resources, J.A.; data curation, J.A.; writing—original draft preparation, K.S.; writing—review and editing, J.A., A.S. and L.M.; visualization, K.S. and Z.S. (Zhassulan Smanov); supervision, J.A.; project administration, L.M. and Y.G.; funding acquisition, L.M. and Y.G. All authors have read and agreed to the published version of the manuscript.

**Funding:** This research was funded by the Regional Collaborative Innovation Project of Xinjiang Uygur Autonomous Region of China (2020E01013); the LU JIAXI International team program supported by the K.C. Wong Education Foundation (GJTD-2020-14), and the High-Level Training Project of Xinjiang Institute of Ecology and Geography, Chinese Academy of Sciences (E050030101).

**Institutional Review Board Statement:** Not applicable.

**Informed Consent Statement:** Not applicable.

**Data Availability Statement:** Not applicable.

**Acknowledgments:** We thank the anonymous reviewers for their valuable comments and suggestions.

**Conflicts of Interest:** The authors declare no conflict of interest.

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
