# Peer review of "Dimensionality-Transformed Remote Sensing Data Application to Map Soil Salinization at Lowlands of the Syr Darya River"

_sustainability, doi:10.3390/su142416696_

Round 1

Reviewer 1 Report

This paper had done a lot of word to soil sampling, soil analysis and combined with remote sensing imagerys to map soil salinization, in addition four modelling methods were used. Generally, this paper's novelty is good, however, the structure is too bad. Strongly suggest authors to change their structure and redisplayed your work in a well-organized structure. After these modification, this paper could be publish in this journal.

In addition, references are a little old. Please add several references at the year of 2021 and 2022.

Title: this title is too general. Soil salinization at agricultural lands, what's your core purpose and key methods you want to display. After reading, i think your key thought was that map the soil salinization of this study area integrated with remote sensing imagerys. Hence, suggest authors to change your title, highlighting your key novelty and key purpose.

Abstract: please also conclude your key thought and key idea, key results and findings. Give reason why these findings could be useful for land management.

Keywords: climate change is not suitable for one keyword.

Line 77: it should be 29

Rearrange your introduction, existing description seems like a experimental report, not like an academic manuscript. Meanwhile, don't divide into several parts in introudction part. More focus should be put on the background, main meaning, what methods that exising studies had used, Is there have some shortcomings, etc.

Section materials and methods:

In this section, you should add one section of 2.2 to introduce the main data you have used or collected. And displayed these data by one table. Not describe these data by poor description. This is like a experimental report. You should give the key information about the methods and data you used. In addition, a technique flowchart is necessary. Please add one flowchart of your work.

Table 1 should be in result part, not in methods part.

in figure 7, 8, 9, have they past the test? What is the P value.

Discussion and conclusion part is too bad. Please rewritten them. Especially for conclusion part, I have no idea why you write these content. Conclusion part should mention the key findings you have acquired not these irrelevant description.

Author Response

We thank the reviewer for the valuable suggestions and submit our replies:

  1. Title - changed it to "Dimensionality-transformed remote sensing data application to map soil salinization at lowlands of the Syr Darya river" to delineate the content and attract interest.
  2. Abstract - added "PCA-transformation of the Landsat-8 satellite images helped to remove redundant spectral information from multiband datasets and to map soil salinity more precisely. This approach simultaneously extends mapping opportunities involving visible and invisible bands and results in a smaller dataset". 
  3. Keywords - added "heating; aridization".
  4. Line 77: we did not fully understand, but in terms of literature, it is entitled as the number 29.
  5. Introduction - rearranged and rewritten with a focus on the significant background of the study.
  6. Section 2.2 - the collected data is included in item 1 (Table 1).
  7. The flowchart is included in Materials and methods.
  8. The results of the statistical tests are in the plots (figures 8,9,10).
  9. Discussion and conclusion part were rewritten.

Reviewer 2 Report

Manuscript entitled “Soil salinization at agricultural lands: the case of the Syr Darya river’s lowlands” by Samarkhanov and team.

Authors highlighted the problem of saving natural (soil) resources and their reclamation measures under current climate change conditions attracts the world community’s close attention.

Abstract: Well written, overall recommendation is missing

Introduction: written well, need updated literature mainly in relation to climate change with evidence.

M & M and Results: Excellent,

Discussion:  Need major revision, it is very generalized need scientific discussion. Pl ass relevant literatures.

Conclusion: Excellent  

Overall, the manuscript has very information. Only need modification in discussion section, it is recommended for further process. 

Author Response

  1. Introduction: updated and considered climate change issues.
  2. Discussion: revised.

Reviewer 3 Report

This study estimated the soil salinity using Landsat-8 imagery and the K-nearest neighbourhood model between 2013-2018 in the Syr Darya river’s lowlands. It also studied the relationship between climate change patterns in these years and soil salinity in the study area. The topic is interesting and has significant implications for soil conservation and sustainable agricultural production. However, the research background and research background have not been clearly indicated in the introduction. More literature needs to be reviewed to highlight what is the scientific research gap. In the methodology, the methods used have not been introduced clearly. For example, how the satellite imageries were processed, the theory of the model used in this study and how the relationship between climate change and soil salinity area was analysed. The results need to be condensed. In the discussion, it is necessary to discuss the practical implications of this study.

Thus, I recommend a major revision before accepting for publication. Moreover, the paper needs to be carefully revised according to the following detailed suggestions.

Line 8: The abbreviate needs to be introduced at the first time used.

Line59-60: This sentence is unclear. soil or salt?

Line 89-90: for calibrating satellite imagery? This sentence is confusing. More details or information on the literature are required.

Line 94: at the top 20cm?

Introduction: There are substantial information on the causes and the impact of soil salinity, but very limited information on how soil salinity has been studied and monitored in the literature and what research gap this paper is bridging.

The following reference might be useful for you to develop a strong background.

Zhao, D., Wang, J., Zhao, X., & Triantafilis, J. (2022). Clay content mapping and uncertainty estimation using weighted model averaging. Catena, 209, 105791.

Zare, E., Huang, J., Santos, F. M., & Triantafilis, J. (2015). Mapping salinity in three dimensions using a DUALEM421 and electromagnetic inversion software. Soil Science Society of America Journal, 79(6), 1729-1740.

Zhao, D., Li, N., Zare, E., Wang, J., & Triantafilis, J. (2020). Mapping cation exchange capacity using a quasi-3d joint inversion of EM38 and EM31 data. Soil and Tillage Research200, 104618.

Line 152: This is unnecessary information.

Line 196:a  is MLR machine learning algorithm? please check.

Line 267: How the relationship between soil salinity and climate change was studied has not been clearly introduced.

Line 261-265: It is very difficult to conclude that clay content is the largest at the top layer by comparing these ranges. The median or mean values of clay content might be a better choice.

Figure 5: According to the triangle, the soil profiles are quite uniform.

Table 3: Please add another column to indicate the depth.

Line 341-347: repeating information that has been indicated in the methodology.

Figure 9: Is this performed on the validation dataset?

Line 405: How the salt-affected area is calculated needs to be introduced in the methodology.

Line 412-415: This sentence is difficult to understand.

Line 418: [45,47] you do not need references to show your research results.

Line 431-436: More information on the implications of the proposed method is required.

Author Response

  1. Introduction revised, remote sensing methods and climate change effects on soil salinity considered. Focus on the PCA method is shown. New literature added. Trend decomposition included.
  2. An abbreviation of the K-nearest neighborhood (KNN) is added to the abstract.
  3. Line 59-60 in the previous version was at the end of the background and have now been replaced by the sentence, "This study utilized TCT images and climate variable trends to assess the salinization of soils in the Kazakh part of the Syr Darya river basin.".
  4. Line 89-90 in the previous version has now been replaced by the sentence, "Many studies described the use of optical remote sensing data to assess or map soil salinity in bare soils.".
  5. Line 94 in the previous version does not contain the soil layer depth information, or we did not correctly understand the question. But most researchers consider the application of Remote sensing to the top soil layer (0 cm).
  6. Recommended literature considered and added to the text.
  7. Line 152: This is unnecessary information - replaced.
  8. Line 196:a  is MLR machine learning algorithm? please check.

  9. Line 267: The relationship between soil salinity and climate change was studied by comparison of their temporal plot patterns.

  10. Line 261-265:  The median or mean value of clay content is also shown.

  11. The figure of the soil texture triangle demonstrates the number of available samples that don't necessarily imply the big difference between their properties, the only difference is that soil moisture with silt content typically showed an increase with the depth. 

  12. Table 3: Adjusted the table's name, "Total share salt chemical composition (major ions) for 3 soil layers", because the idea was to demonstrate the general content for total samples, and deepening doesn't mean a big difference.

  13. Line 341-347: Removed repeating information indicated in the methodology.

  14. Figure 9 (10) Machine learning was applied to the validation dataset.

  15. Line 405: Salt-affected area calculation compared to climate data included in the methodology.

  16. Line 412-415: This sentence is difficult to understand, Line 418: [45,47] you do not need references to show your research results, Line 431-436: More information on the implications of the proposed method is required. - Discussion and Conclusion are adjusted.

Round 2

Reviewer 1 Report

All authors have well addressed all my concerns. I suggest this manuscript could be accepted for publication.

Reviewer 3 Report

N/A